# Experimental Protocol to Assess Neuromuscular Plasticity Induced by an Exoskeleton Training Session

**DOI:** 10.3390/mps4030048

**Published:** 2021-07-13

**Authors:** Roberto Di Marco, Maria Rubega, Olive Lennon, Emanuela Formaggio, Ngadhnjim Sutaj, Giacomo Dazzi, Chiara Venturin, Ilenia Bonini, Rupert Ortner, Humberto Antonio Cerrel Bazo, Luca Tonin, Stefano Tortora, Stefano Masiero, Alessandra Del Felice

**Affiliations:** 1Department of Neurosciences, Section of Rehabilitation, University of Padova, via Belzoni, 160, 35121 Padova, Italy; emanuela.formaggio@unipd.it (E.F.); giacomo.dazzi@studenti.unipd.it (G.D.); chiara.venturin@studenti.unipd.it (C.V.); stef.masiero@unipd.it (S.M.); alessandra.delfelice@unipd.it (A.D.F.); 2School of Public Health, Physiotherapy and Sports Science, University College Dublin, 4 Dublin, Ireland; olive.lennon@ucd.ie; 3g.tec Medical Engineering GmbH, 4521 Schiedlberg, Austria; jimmy_sutaj@hotmail.com (N.S.); ortner@gtec.at (R.O.); 4Ospedale Riabilitativo di Alta Specializzazione di Motta di Livenza, 31045 Treviso, Italy; ilenia.bonini.ib@gmail.com (I.B.); HumbertoAntonio.CerrelBazo@ospedalemotta.it (H.A.C.B.); 5Department of Information Engineering, University of Padova, 35131 Padova, Italy; luca.tonin@unipd.it (L.T.); stefano.tortora@unipd.it (S.T.); 6Padova Neuroscience Center, University of Padova, 35129 Padova, Italy

**Keywords:** rehabilitation, stroke, aging, EEG, EMG, neuromuscular plasticity, exoskeleton

## Abstract

Exoskeleton gait rehabilitation is an emerging area of research, with potential applications in the elderly and in people with central nervous system lesions, e.g., stroke, traumatic brain/spinal cord injury. However, adaptability of such technologies to the user is still an unmet goal. Despite important technological advances, these robotic systems still lack the fine tuning necessary to adapt to the physiological modification of the user and are not yet capable of a proper human-machine interaction. Interfaces based on physiological signals, e.g., recorded by electroencephalography (EEG) and/or electromyography (EMG), could contribute to solving this technological challenge. This protocol aims to: (1) quantify neuro-muscular plasticity induced by a single training session with a robotic exoskeleton on post-stroke people and on a group of age and sex-matched controls; (2) test the feasibility of predicting lower limb motor trajectory from physiological signals for future use as control signal for the robot. An active exoskeleton that can be set in full mode (i.e., the robot fully replaces and drives the user motion), adaptive mode (i.e., assistance to the user can be tuned according to his/her needs), and free mode (i.e., the robot completely follows the user movements) will be used. Participants will undergo a preparation session, i.e., EMG sensors and EEG cap placement and inertial sensors attachment to measure, respectively, muscular and cortical activity, and motion. They will then be asked to walk in a 15 m corridor: (i) self-paced without the exoskeleton (pre-training session); (ii) wearing the exoskeleton and walking with the three modes of use; (iii) self-paced without the exoskeleton (post-training session). From this dataset, we will: (1) quantitatively estimate short-term neuroplasticity of brain connectivity in chronic stroke survivors after a single session of gait training; (2) compare muscle activation patterns during exoskeleton-gait between stroke survivors and age and sex-matched controls; and (3) perform a feasibility analysis on the use of physiological signals to decode gait intentions.

## 1. Introduction

Stroke has a high personal and societal burden. It is the second most common cause of death (WHO fact sheet 2017) and a leading cause of adult physical disability [1], affecting 17 million people worldwide each year. Demographic trends of an ageing population and escalation of risk factors will lead to an estimated 32% increase in DALYs (disability adjusted life years) lost from 2015 to 2035 (609,721 to 861,878) (WHO fact sheet 2017).

Recovery after stroke is often incomplete with poor outcomes commonly reported. Improving recovery and long-term outcomes after stroke has become both a clinical and scientific challenge [2]. Despite conventional gait rehabilitation enhancing walking velocity, endurance [3], and balance, especially during the sub-acute phase [4], it can be physically onerous for therapists and particularly challenging to facilitate an effective gait pattern for motor learning, in terms of cadence, inter-limb coordination, and muscle timing [5]. From this perspective, exoskeleton rehabilitation of overground walking has the potential to improve the functional recovery of gait by allowing repetitive task practice with less therapist effort [6]. Exoskeleton uptake in rehabilitative clinical practice is slowly gaining momentum [6,7,8,9], but its adaptability to real-time and ongoing modifications induced by the rehabilitation itself remains an unmet goal. Exoskeletons, as currently deployed with prescribed gait trajectories and pre-defined assistance levels, generate poor human-machine interaction, especially in individuals that retain a degree of voluntary motor capability.

Past research has explored the possibility of exploiting non-invasive, electroencephalography (EEG)-driven interfaces to operate exoskeletons [10]. To date, the number of published works involving healthy users remains a single-digit figure [11]. While a few studies have reported a significant and positive effect when using EEG-driven exoskeleton gait training for people affected by chronic spinal cord injury [12,13], no studies reported EEG-driven exoskeleton gait operated by a clinical population such as stroke where the pathology is at brain level [6].

In most cases exogenous and discrete paradigms (e.g., steady state evoked potentials or P300) have been exploited to trigger predefined walking patterns of an exoskeleton [11]. This approach strongly limits the human-machine interactions as well as the active involvement of motor networks. To date, only a few studies reported the possibility to decode lower limb joint kinematics and walking patterns via EEG [14,15].

To spearhead a paradigm change in the use of exoskeletons in rehabilitation after stroke and address pressing healthcare needs, this protocol aims at overcoming current limitations in exoskeleton adoption in clinical practice by collecting data to investigate:Neurophysiological signals as biomarkers of brain plasticity induced by exoskeleton training. EEG and/or electromyographic (EMG) pattern modifications induced by robotic training may be markers to quantify and track neuromuscular plasticity [6,9]. Since commonly used clinical scales rely mainly on subjective functional assessments and are not able to provide a complete description of patients’ neuro-biomechanical status, current clinical tests should be integrated with specific physiological measurements to obtain a deeper understanding of the effect of the rehabilitative intervention [16].Robotic training to restore gait patterns. Conventional rehabilitative interventions using robotic technology facilitate sensorimotor recovery by supporting and motivating participants to practice specific tasks [17,18]. Exoskeletons are a promising task-oriented tool intended to restore a more physiological gait pattern in highly compromised individuals [6,7,8] and to recover physiological alternated activations of lower limb muscles [19].Neurophysiological signals as predictors of gait trajectory. EEG and EMG signals related to gait in healthy older adults and stroke survivors are attracting researchers’ attention [20,21]. Concurrent analysis of these physiological signals may deepen the understanding of the neuro-motor control of walking [22,23]. We need to single out a clear-cut biomarker that precedes biomechanical modifications and which is most likely to be a cerebral signature, i.e., specific anticipatory potentials [24].

## 2. Materials and Methods

The protocol is part of the PROGAIT project [25]. The study will enroll stroke survivors and sex- and age-matched controls recruited at the High Specialization Rehabilitative Hospital (ORAS—Ospedale Riabilitativo ad Alta Specializzazione), Motta di Livenza (Italy) (see inclusion criteria and clinical data section). Following familiarisation with the device, participants will undergo a single training session with an active exoskeleton for gait rehabilitation, e.g., EKSO GT (EKSObionics Inc., Richmond, CA, USA). The resting state cortical activity and the gait characteristics pre- and post-training session will be recorded through EEG, EMG and inertial measurement units (IMU)—see Materials and Procedure Sections. Cortical and muscular activity and gait characteristics will be recorded also during the EKSO gait training session (see Figure 1). Trained expert personnel will guarantee both participants’ comfort and good data quality (i.e., physicians for participants selection, enrollment, clinical data collection, and supervision of data collection sessions; certified physical therapists expert in robotic rehabilitation of gait; biomedical engineers to ensure data quality during data collection and to process data).

### 2.1. Inclusion Criteria & Clinical Data

#### 2.1.1. Control Participants

Inclusion criteria:sample size: N ≥ 10 participants;50% female (F), 50% male (M) participants;40–60 years-old.

Exclusion criteria:previous or ongoing orthopedic or rheumatological disease;central and peripheral nervous system diseases;dermatological problems or anthropometric measurements limiting exoskeleton use;intake of any drug with a central nervous system effect;scalp lesions.

#### 2.1.2. Stroke Survivors

Inclusion criteria:sample size: N ≥ 20;50% F, 50% M;(40–60) years-old;Within 3 months from stroke;First ever supratentorial ischemic stroke (lesion localization will be collected and considered as a covariate for the statistical analyses);FAC >2 [26], as stroke survivors are required to perform a free overground walking trial before and after the EKSO training session, as well as the EKSO walking in free mode (i.e., with motors off).

Exclusion criteria:receptive aphasia;unilateral spatial neglect or cognitive issues that may limit capacity for informed consent or full cooperation;co-existing orthopaedic, dermatological or neuromuscular disorders that limit EMG data testing;previous history of EEG abnormalities or epilepsy;scalp lesions (breaches).

#### 2.1.3. Clinical Data to Collect from Stroke Survivors


Drugs intake;Medical therapy;Date of the clinical event;Lesion location (Bamford classification [27]);Intervention (e.g., thrombolysis, thrombectomy or standard medical care);Neuroimaging data (Magnetic Resonance Imaging and/or Computed Tomography);National Institutes of Health Stroke Scale to assess the impairment caused by the stroke (NIHSS) [28];Medical therapy;Oxford Cognitive Screen, which was developed to identify common deficits after stroke, such as aphasia, spatial neglect, apraxia, and reading/writing problems [29];Muscle strength assessed with the Medical Research Council (MRC) Scale [30] of ileopasoas, quadriceps femoris, hamstrings, tibialis anterior, triceps surae;Sensory testing: light touch, pain, proprioception and vibration;Modified Ashworth Scale of the affected lower Limb (MAS): to evaluate the muscle spasticity of the hip adductor, knee extensor, and ankle plantar flexor [31];Modified Rankin scale of Disability (mRS) to measures the degree of disability in the daily activities [32,33];Functional Ambulation Category (FAC) to evaluate functional ambulation [26], collected retrospectively (i.e., before) and after the acute event;Modified Barthel Index (mBI) to assess independence in activities of daily living [34];Orpington Prognostic Score as assessment of stroke severity, e.g., motor deficits, proprioception, motion and cognition [35].


#### 2.1.4. EKSO Inclusion Criteria


Subjects must tolerate standing;Height: 1.6–1.9 m;Body mass <100 kg;Upper leg length discrepancy ≤1.3 cm;Lower leg length discrepancy ≤1.9 cm;Muscle hypertonia quantified as MAS ≤2 at lower limbs [31];No muscle-tendons retractions.


### 2.2. Materials

*EEG wireless system*, e.g., g.NAUTILUS PRO system with 64 electrodes (g.tec medical engineering, Schiedlberg, Austria).

*EMG and IMU wireless system*, e.g., Cometa MiniWave Waterproof EMG sensors complemented with WaveTrack Waterproof, consisting of a 3D accelerometer, a 3D gyroscope and a 3D magnetometer (Cometa srl, Milan, Italy).

*EKSO GT*${}^{©}$ (EKSObionics Inc., Richmond, CA, USA). The device consists of two lower limb supports connected by a torso structure. The torso, which is worn like a back-pack, contains the device’s batteries and electronic control system. The torso component also features handles for a therapist to provide support during movement. The legs of the device are coupled to the patient’s legs using straps at the thigh, shins and feet. An abdominal binder and straps around the torso and the shoulders secure the upper body to the device. Electric motor actuators at the hip and knee joints of the legs allow range of motion in the sagittal plane. All other planes of motion are locked at the hip and knee. The ankle joints of the device are locked in all degrees of freedom, but are highly sprung in the sagittal plane, which allows flexibility for sitting, standing and walking and reduces the complexity and weight of the device [36].

EKSO GT is an active exoskeleton for overground gait rehabilitation that can be used in different modes, adapted to the user’s needs. EKSO can be tuned for step actuation and amount of power to be supplied [37]. The EKSO steps can be actuated by: (i) the physiotherapist by pushing a button (FirstStep™mode); (ii) the patient by pushing buttons, either on the crutches or the walker (ActiveStep™mode); (iii) the patient by shifting body weight laterally and then forward to a predefined threshold (ProStep™mode); and (iv) the patient by only shifting body weight laterally to a predefined threshold (ProStep Plus™mode). The power supplied to each leg during stepping can be adjusted based on three levels. Specifically, EKSO actuators can provide: (i) full power to both legs, i.e., leg trajectory is fully driven by the robot and no muscle strength is required to the user (Bilateral Max Assist); (ii) the additional power needed to complete a correct and smooth leg trajectory, complementing the user leg strength (Adaptive Assist); and (iii) a fixed and predefined amount of power to help the user completing a correct and smooth leg trajectory within a predefined time window (Fixed Assist). All these modalities call for the user to have proper control of balance and to be able to shift the body weight laterally, with the only assistance for such tasks coming from crutches/walkers and the physiotherapist. A further setting for the EKSO allows the robot to completely follow the user movements, without applying any supplemental power other than supporting its own weight (Free mode).

Before use, EKSO calls for a customization session performed by a trained physical therapist that will set the parameters based on participants’ anthropometry (Table 1).

### 2.3. Procedure

The study will be conducted according to the guidelines of the Declaration of Helsinki. Full ethical approval has been received from the Institutional Review Board (Treviso, PROGAIT TRAINER, 24.10.19) and written informed consent will be provided by all participants.

#### 2.3.1. Data Acquisition

Wireless high-density EEG recordings will be acquired. Electrode-skin impedance will be maintained <40 kΩ. An expert researcher will identify the position of Cz based on the 10–20 measurements and gently scrub the scalp before placing the EEG cap and the gel. The gel will be applied starting from the reference electrode (on the ear lobe), then the ground electrode, Cz and all the other channels.

In parallel, a wireless system consisting of EMG probes and IMUs will be used to record both muscle activity signals and kinematic data synchronized with EEG recordings. EMG recordings will be acquired bilaterally from vastus lateralis, biceps femoris, tibialis anterior and gastrocnemius lateral head.

Skin preparation will be performed by removing dead skin cells (i.e., gentle abrasion) and moisturization of the skin, in case of excessively dry skin, or cleaning skin with an alcoholic wipe, in case of oily skin [38]. The electrode-skin impedance will be maintained <50 kΩ. The recordings will be sampled at 2 kHz. An expert researcher will select the position of the electrodes to minimize crosstalk effects [39] as suggested by [38,40] (Table 2).

Accelerometer and gyroscope data will be sampled at 250 Hz. One IMU will be secured on the lower back (at the fifth lumbar vertebra, L5) with a double-sided adhesive tape to measure the movements of the pelvis and thus track the movements of the Center of Mass (CoM) [41,42,43] and detect any turning activity by observing the integral of the gyroscopic vertical component [44]. Two IMUs will be attached on the lateral aspect of both shanks to detect foot-strike and foot-off events of the gait cycle, as in [45].

EEG, IMU and EMG data will be streamed into a custom-made software created in C# and using the g.NEEDACCESS .NET API (g.tec medical engineering, Schiedlberg, Austria) combined with the COMETA .NET API (Cometa srl, Milan, Italy), which ensures synchronization among the collected signals. Prior to the recordings, synchronicity of the system was verified by feeding test-signals and test triggers in parallel to the sensors. The analysis showed a delay, on average, of 4 ms at the start of an acquisition and a drift of about 26 ms during an acquisition period of 30 min. That drift is generated by the two clocks in the EEG and EMG system, which do not run in perfect synchrony.

EEG and IMU data will be originally collected at a sampling rate of 250 Hz, whereas EMG data are collected at 2000 Hz. To ensure data synchronization among the three signals, EEG and IMU data will be linearly upsampled to the lowest common multiple (i.e., 2000 Hz). When the data collection command is armed, the software will start reading each channel from the EEG cap and EMG-IMU receivers. Data streaming to the custom-made software will not start simultaneously, and initial data gathered will be discarded until all channels providing real-time inputs are fully synchronised.

#### 2.3.2. Experimental Protocol

Before starting recordings, EEG, EMG and IMUs signals will be qualitatively checked by expert researchers. If needed, sensor placements will be adjusted. Impedance values will be recorded and saved at the beginning and at the end of the experimental protocol. The whole protocol will last approximately 100 min and will be repeated once. Before collecting the data during the EKSO walking, participants will familiarize with the robot to ensure the settings are appropriate to their needs and to not record transients in the neurophysiological signals. The full instrumentation set will be employed during the whole protocol: resting state, free-walking pre- and post-training session with the EKSO and EKSO-walking.

Resting state recording. After the signal quality check, a 5 min EEG resting state recording will be acquired, asking the participant to sit down in a dimly lit sound-attenuated room with her/his eyes open (fixing a point in front of her/him) and slightly open the mouth to relax facial muscles.

Free-walking recording. Participants will be asked to walk at a self-selected pace for at least 30 m (i.e., one trial: 15 m to go, turn around, and walk back another 15 m, see Figure 2a). The ”start walking” command will be given at least 3 s after the start of recording, with the participant standing as still as possible. A roll-pitch correction will be estimated from this condition to correct for possible misalignment of the sensor axes with the global anterior-posterior (AP), medial-lateral (ML) and vertical (VT) directions of the body [46]. Participants will be allowed to walk with walking aids when required. Where a participant with stroke is unable to perform the free walking task due to motor impairments, this step will be skipped and the reason documented.

During exoskeleton gait tasks, the IMU previously placed on the L5 vertebra during free walking conditions will be moved to the equivalent location on the device, approximating to the L5 lumbar region.

EKSO walking session. Participants will be fitted with the EKSO device, which will be programmed to the ProStep™mode with three modes of supplied power (10 min each): Bilateral Max Assist mode (M), Adaptive Assist mode (A), Free mode (F). The EKSO parameters (e.g., swing velocity, cadence, lateral shift) will be adjusted by the chartered physiotherapist certified in EKSO use as a function of each user’s needs. These parameters will then remain standardised throughout all subsequent test procedures After each mode is set, participants will be asked to perform a few walking trials to ensure any adaptation effect is over. For safety reasons, EKSO walking will always be performed with the use of walking aids. Data acquisition will then start and the session will proceed as the Free-walking recording, with the 3 s of standing before starting walking. At the end of the session, the IMU on the lower back of the robot will be placed again on participants’ L5 (same position of the first Free walking trial).

Free-walking recording. When the EKSO device had been removed and the IMU is reinstated at the L5 anatomical region, three further walking trials (Figure 2a) will be collected to observe whether there is a training effect or re-establishment of the baseline gait pattern. If this step is not possible with stroke survivors (either for fatigue or motor impairment), it will be skipped and again reasons will be documented.

Free-walking recording. When the EKSO device had been removed and the IMU is reinstated at the L5 anatomical region, three further walking trials (Figure 2a) will be collected to observe whether there is a training effect or re-establishment of the baseline gait pattern. If this step is not possible with stroke survivors (either for fatigue or motor impairment), it will be skipped and again reasons will be documented.

Resting state recording. After having removed both EMG and IMU sensors, 5 min EEG resting state recording will be acquired in the same condition as the first recording.

### 2.4. System Usability and Wearability

Subjects will be instructed not to talk during free walking and EKSO walking trials. After each trial, the BORG CR-10 scale will be administered to evaluate perception and experience (including pain and exertion) associated with the use of the EKSO [47]. The BORG CR-10 will be repeated for each mode of use of the EKSO. The wearability and acceptance of the device will be assessed using the Comfort Rating Scale (a 21-point scale developed specifically to assess the comfort of wearable technologies) [48,49] and the System Usability Scale (10 items to be scored on a 5-point likert scale basis) [50].

### 2.5. Foreseen Data Processing

The primary aim of this protocol is to study the short-term plasticity induced by a session of gait training with an exoskeleton for overground walking. Toward this aim, we plan to study the EEG topographies in the resting state pre- and post-training through microstates analysis [51] to evaluate the most representative topography pre- and post-training. Building on this knowledge and considering the stroke survivors T1-weighted MRI sequence, we will study the source cortical waveforms [52] that are active (i.e., source waveforms with higher amplitudes) and their frequency content (power spectra in the canonical EEG frequency bands) pre- and post-training.

During gait, if the signal-to-noise ratio will be acceptable, we aim to apply time-varying effective connectivity algorithms to estimate the active brain regions and their interactions during gait with and without the use of the exoskeleton in parallel to the analysis of the gait parameters (e.g., foot-strike and foot-off events, gait symmetry and complexity) [42,45,53]. For instance, the temporal and spectral EEG analysis time-locked with each step onset may reveal the existence of bio-markers that could be exploited as reliable predictors of the lower limb motion, such as the movement related cortical potential (MRCP), i.e., a negative deflection in the low EEG frequencies (0.1–3 Hz) before the movement onset [54]. We will analyze the effect of the robot-assisted gait-training on the characteristics of these brain potentials. The negative amplitude of the MRCP has been found to be correlated with the level of engagement of the participant in performing the motor task [55].

In parallel to the EEG features, EMG data will be processed to calculate muscle synergies [56,57], signal amplitudes (RMS) and center of activity throughout the gait cycle (CoA) [58]. The CoA, which is the first trigonometric moment of the signal distribution, has been previously used to compare muscle activity and timing over different conditions [58,59]. Shin-mounted IMUs will serve as a tool to detect foot-strike and foot-off of gait using validated algorithms for gait event detection [45]. The waist-mounted IMU will rather be used to compute indices of overall gait symmetry [42] and complexity [53], which we expect to vary between pre- and post-training sessions.

Eventually, results from stroke survivors and age and sex-matched participants will be compared through statistical analyses.

## 3. Expected Results

The key innovation of the proposed protocol is in the co-registration of data from three synchronised systems (high-density EEG, surface EMG and IMU) before, during and after robot-assisted gait-training to try to answer the following needs:Investigate how a robotic device interacts with the end-user in able bodied subjects and individuals with neurological disorders;Define the neurophysiological processes underlying both locomotor control with and without an exoskeleton and gait recovery through a robot-assisted gait training;Determine optimal training parameters for individualized gait training (based on knowledge built on 1. and 2.).Boost the clinical use of EEG, EMG and exoskeletons for gait rehabilitation in clinical settings both for clinical and research purposes and help breaking down barriers associated with cultural background and lack of expertise of the multi-disciplinary teams, which are crucial for this kind of intervention [6,60,61];Determine the feasibility of controlling an exoskeleton with physiological (e.g., EEG and/or EMG) signals.

To achieve these aims, we need to study not only kinematics and clinical scales before, during and after exoskeleton-training as previously reported in [37,62,63], but critically the changes that occur in corresponding brain signals. Previous research that has focused on cortical activity during walking has been limited by technical constraints, e.g., the EEG was recorded either without recording synchronised muscle activity [15], or only healthy subjects [64,65], or using a limited number of EEG electrodes instead of a high-density EEG [66], or separately from EMG, kinematics and clinical scales [67]. In addition, the limited number of studies synchronizing EEG, EMG and kinematics recordings during overground walking with and without an exoskeleton are mainly conducted in able-bodied subjects and on treadmills [68]. To specifically overcome these technical and study design limitations and answer to our research questions:A high-density EEG set-up is required to reconstruct the brain cortical activity minimizing leakage effects, targeting the brain areas involved in the tasks (e.g., through power spectra or time-frequency analysis) and their interactions (e.g., effective connectivity and network/graph analysis [69,70,71])The synchronous kinematics is required to detect gait phases to segment the EEG and EMG signals according to foot-strike and foot-off events and quantify gait symmetry and complexity [45,53]The synchronous EMG signals are required to identify antagonistic-agonistic muscular activation during gait and muscles response to exoskeleton.

Figure 3 reports an example of EEG signals pre- and post-training in a control subject. Qualitatively observing Figure 3, the average EEG activity changes after training in both time (first row) and frequency (second row) domain. The spectral power density in Alpha (i.e., (8–12) Hz) and Beta (i.e., (13–30) Hz) bands increases after training. Since motor impairment after stroke is a leading cause of disability and Alpha and Beta rhythms (i.e., (10–30) Hz) are the cortical rhythms mainly involved in motor planning and control [72], these preliminary results support the possibility to promote cortical activity in higher frequency rhythms (i.e., >8 Hz) in the sensorimotor areas after EKSO walking.

Figure 4 reports an example of changes elicited in the activity of the muscles in the lower limb in an healthy individual before and after the robotic training session. During the free-walking recorded before the training session, the timing of activation of all the recorded muscles follows the expected regular and stride-synchronous pattern [73]. This physiological pattern is modified after the use of the EKSO. Qualitatively inspecting Figure 4, the more evident changes with the pattern observed pre-training are: (i) less regular timing of foot-strike and foot-off events between pre- and post-training session; (ii) muscles generally show a prolonged activity during the free-walking after the training session; and (iii) antagonist muscles show a larger co-contraction with consequent stiffening of lower limb joints during the free walking post-training. Such stiffening action is also visible when inspecting the sagittal angular velocity of the shank, which is larger and regular before the training session and reduced and less smooth during the free-walking after the use of the EKSO, with the pattern progressively changing from the “EKSO-walking type” to the free walking pre-training (Figure 4, last two rows). Although the described patterns seem to highlight a worsening in the efficiency of gait management (i.e., agonist-antagonist muscles are in co-contraction rather than in counter-phase), it is worth noticing that mechanisms underlying the changes induced by a robot for overground gait-training are still unclear. It has been hypothesized that the structure of the robot itself (i.e., no mobility and unpowered ankle joint) produces an overload of the muscles in the lower limb of healthy subjects, who also perceive and react to the exoskeleton motion [74]. However, the intrinsic redundancy of neuromotor strategies during walking, which are fundamental to gait adaptability, acts as confounding factor when interpreting EMG data collected before and after a single training session performed with an exoskeleton [58]. It is also worth highlighting that we cannot take for granted that the same changes will be observed in individuals with neurological disorders. Specifically, readjustments towards a paraphysiological timing of muscle activation is observed in post-stroke individuals, but the mechanisms that drive this result are unknown [57,75].

There is evidence of short-term neuroplasticity of brain connectivity in chronic stroke survivors after a single session of gait training. These findings were seen to be magnified when using an exoskeleton device in comparison to standard training programs [76] and are expected in both able-bodied participants and stroke survivors.

Central nervous system reorganization can also be reflected by changes in muscle activation patterns. Exoskeleton gait training post-stroke has previously been shown to increase the number of muscle synergy modules describing activations during gait [77]; in addition to improvements in volitional control and muscle activation timing [75]. We expect similar findings to emerge in our stroke population in the current study, while we expect an effect of the training on the activity recorded over the sensorimotor area that is worth investigating also in healthy controls to better understand the human-machine interaction.

The proposed analysis has utility for future studies where it can be applied to data collected in stroke survivors to track longer-term outcomes of functional recovery. Establishing a correlation between neurophysiological signatures of brain and muscle plasticity during gait training after stroke and outcomes of functional gait restoration is a key step towards the identification of prognostic biomarkers of gait recovery. This study with its in depth analysis of the ongoing modulations of brain and muscle signal during robotic gait training will provide new insight into human-machine interaction. Very little is known on this topic, and the existing knowledge gap has prevented a fine-tuning and a personalization of robotic training and rehabilitation up to now. This gap in knowledge is also due to the difficulties to quantify the relative contribution of movement artifacts on EEG signals during gait with and without the exoskeleton. Several methods have been proposed in the literature to separate non-brain components in locomotion tasks through offline analysis, such as blind source separation or artifacts subspace reconstruction [78,79], that allowed to reveal cortical modulations during walking [22,65,80]. However, only a few studies faced the problem of rejecting gait-related artifacts in real-time [81,82], and none of them considered the interference introduced by the exoskeleton support. The design of a real-time denoising framework from the data collected represents a fundamental step for the design of a closed-loop system for walking rehabilitation and assistance driven by brain activity.

The identification of the movement intention of the user from EEG signals, that does not always correspond to the motor outputs (i.e., in stroke survivors) will provide information for future developments of a closed-loop system, integrating this information. This foundational work will pave the way for next generation robotic devices for rehabilitation and assistive use. The joint acquisition of EEG and EMG data will open to novel hybrid solutions for human-exoskeleton interfacing. The combination of EEG and EMG classification may improve the recognition of motion intent [83,84], e.g., by exploiting the spectral and/or temporal relationship between brain and muscle activity during walking to enrich the prediction that would be obtained with the two signals singularly [85].

This study which aims to collect and integrates clinical and neurophysiological data will provide vital insights on feasible strategies to fine-tune the robotic device parameters to produce the best neuromuscular pattern. Fine-tuning and personalising robotic adaptive modes may: (1) improve patients’ recovery during a period of heightened plasticity, i.e., up to 3 months after the acute event; and (2) propose a customized rehabilitation training (with the ekso) to induce a long-term plasticity in the patient group.

## Figures and Tables

**Figure 1 mps-04-00048-f001:**
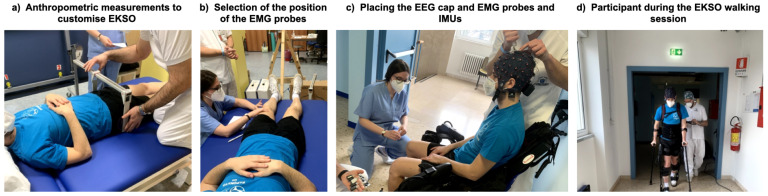
Participant preparation: (**a**) anthropometric measurements for EKSO customisation; (**b**) minimal crosstalk area recognition for EMG sensors placement; (**c**) EEG cap, EMG electrodes and probes and IMUs placement; (**d**) example of EKSO walking.

**Figure 2 mps-04-00048-f002:**
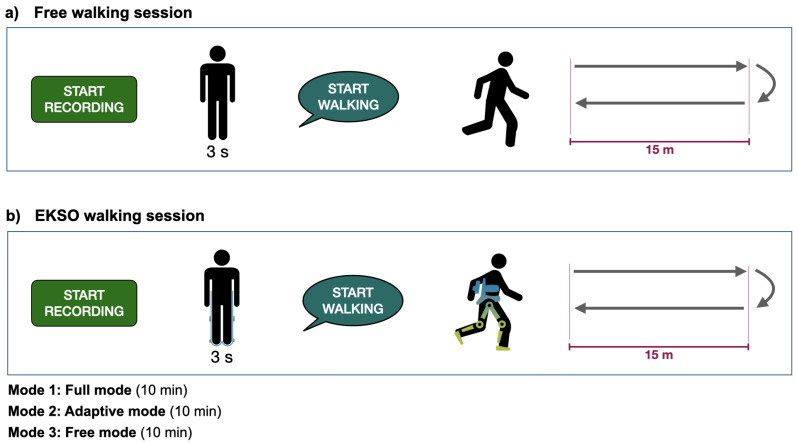
Walking trials: (**a**) Free walking; and (**b**) EKSO walking.

**Figure 3 mps-04-00048-f003:**
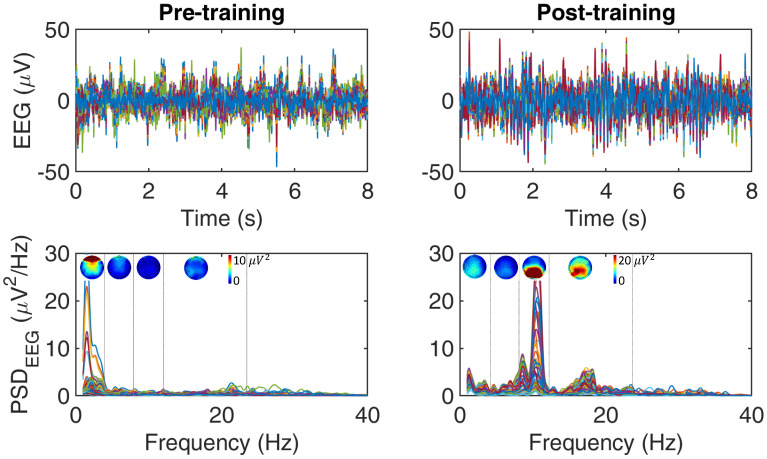
Example of filtered (in [1–40] Hz) EEG time series pre- (first column) and post- (second column) training with EKSO in a control subject. In the first row, it is reported 8 s of the EEG time-series. In the second row, it is displayed the power spectral density (PSD) computed between 1 and 40 Hz of the signals reported above. PSD was computed for the EEG signal recorded in each electrode using Welch’s 50% overlapped 2-s segment averaging estimator. For each EEG frequency band (i.e., Delta (1–4) Hz), Theta (4–8) Hz, Alpha (8–12) Hz, Beta (12–24) Hz), it is also reported the topographic map of the power spectra.

**Figure 4 mps-04-00048-f004:**
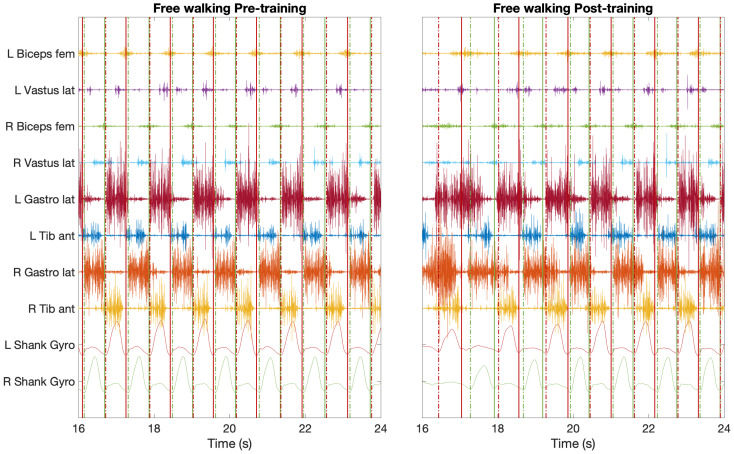
Example of filtered EMG time series during free-walking pre-training and free-walking post-training with the EKSO. Vertical lines correspond to foot-strike (solid lines) and foot-off (dashed) gait events. Red lines highlight the events for the left side, whereas the right side events are reported in green.

**Table 1 mps-04-00048-t001:** Anthropometric measurements needed to customize the EKSO to the participant body.

Anatomical District	Participant Positioning	Measurement
Hip width	Supine (lying down)	Distance between great throcanters
Upper leg (left/right)	Supine (flexed knee)	From gluteus to the top of the flexed knee
Lower leg (left/right)	Sitting	From shoe soles to the top of flexed knee

**Table 2 mps-04-00048-t002:** Guidelines to find minimal crosstalk area (MCA) for EMG electrode placement [38,39].

**General Guidelines**	
**Electrode type**	Kendall ARBO Ref 31.1245.21
**Inter-electrode distance (IED)**	1.5 cm
**Skin preparation**	***Dry skin:***
	· Wet the skin on the MCA
	· Rub the skin to remove dead cells
	· Dry the skin before sticking electrodes
	***Oily/Creamy skin:***
	· Rub the skin on MCAs with alcoholic wipes
	· Rub the skin to remove dead cells
	· Dry the skin before sticking electrodes
**Inter-electrode line**	Longitudinally to muscle fibers
**Starting posture**	
**Vastus lateralis**	Supine, lying down
**Biceps femoris**	Prone, lying down
**Tibialis anterior**	Supine, lying down
**Gastrocnemius lateralis**	Prone, lying down
**Electrode location**	
**Vastus lateralis**	25% on the line between the Gerdy’s prominence and the anterior iliac spine
**Biceps femoris**	Lateral side of back thigh, halfway between ischial tuberosity and lateral epicondyle of the tibia
**Tibialis anterior**	On muscle belly at 25% of the line between head of fibula and lateral malleolus
**Gastrocnemius lateralis**	On muscle belly at 25% of the line between head of fibula and lateral malleolus
**Selectivity check**	
**Vastus lateralis**	With flexed hip, complete knee extension
**Biceps femoris**	With neutral hip flexion, complete knee flexion
**Tibialis anterior**	Pull up the toes. Talus-varus movement to check for minimal crosstalk from peroneus longus
**Gastrocnemius lateralis**	Push toes against resistance

## Data Availability

Not applicable.

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
