# Peer review of "Experimental Protocol to Assess Neuromuscular Plasticity Induced by an Exoskeleton Training Session"

_mps, 2021, doi:10.3390/mps4030048_

Round 1
Reviewer 1 Report
In this work, tha authors propose and describe a protocol for measuring EEG, EMG, and IMU data of control subjects and stroke survivors while walking with and without the help of an exoskeleton. With the final goal of quantifying neuro-muscular changes after the use of the exoskeleton and of usign EEG and/or (this is not clear) EMG to control the exoskeleton. The paper is well written, the topic is a very interesting one but in my opinion the authors should stress more the innovation aspects of their protocol and study. This is a major aspects that is missing in the paper.
I think that the author should highlight more how this protocol is innovative with respect to already published works. For example: recording EEG, EMG and kinematic while walking with and without exo is that new? How are they making it more interesting compared to other works in the litterature?
Perhaps the innovation will be in the parameters the authos are going to extract? if so, I think they should be more specific.
Will the innovation be in how the exo will be controlled by eeg signal? if so I think it is necessary to describe how they are planning to do it. Is the control signal used for the futur control only EEG or a combination of EEG and EMG?
They just listed their three points of innovations but not speculate too much on what and how they are going to achieve them. I think that is a very important thing that is missing.
Minor comments:
Abstract: Results and conclusion are missing. Even if this is mostly about the protocol, I think the authors should try to say something about the expected results
line 57-59 "Neurophysiological signals as biomarkers of brain plasticity induced by exoskeleton training." In this sentence I think it is worth citing also the work of Micera's group that came out last year, wehre they identified biomarkers from kinematic, emg and eeg of subacute stroke patients while training with an upper limb exoskeleton https://doi.org/10.1088/1741-2552/ab9ada
section 2.1.2 is there any inclusion criteria about hemisphere lesion location? like stroke in the right hemisphere only?
line 289 "survivors" it sounds strange to me the word "survivors" at the end of the sentence "in addition to improvements in volitional control and muscle activation timing survivors"
In the section "expected results" they only show EEG data from a control subject, how about EMG data?
Reviewer 2 Report
The manuscript describes a Protocol to Assess Neuromuscular Plasticity pre, post, and during a single session of overground Robot-Assisted Gait Training (o-RAGT) in both healthy and stroke subjects.
The topic is very interesting, and I agree that limited literature has been published on the neurophysiological effects of o-RAGT. Indeed, an assessment before and after a period of o-RAGT (more than one session) will be exciting for the scientific community and I suggest adding it to the discussion.
The protocol is well-written, and I think it can be accepted for being published after the following minor revisions:
- Introduction: the state of the art is clear and complete. Recently, a special issue of Frontiers in Neurology assessed the Barriers Limiting Widespread use of sEMG in Clinical Assessment and Neurorehabilitation, and a paper described the typical limitation in conducting experimental tests for sEMG Assessment During Overground Robot-Assisted Gait Training in Subacute Stroke Patients. I think it could be noteworthy to mention it.
- Materials & Methods:
What’s the primary outcome of the study? What’s the hypothesis of the study?
How did you choose the number of healthy and stroke subjects?
Please write the participants' criteria dividing the inclusion and the exclusion ones.
You considered the “FAC > 2” inclusion criterion as one of the EKSO inclusion criteria. However, the EKSO can be used without any limitations, also in subjects who are not able to walk. Maybe the “FAC > 2” criterion is specific for stroke subjects?
The Ekso GT allows different parameter settings: swing velocity, step length, lateral shift. How did you set them? Are you going to compare data by varying them? Please add more information.
Data synchronization: how did you synchronize the systems? Trigger?
During the Free walking the EKSO walking trials, could the participant use an assistive device (canes, crutches, or a walker)?
How long does the whole experimental test last approximately? Are you going to do it once or do you plan to record 2 or more sessions for each participant? Do the participants familiarize themselves with the EKSO or are you going to record the very first time they’ll use it?
It's not clear if you’re going to record EEG and EMG also during the Free walking the EKSO walking trials. How are you going to manage the wires in a 15m-long pathway?
Please add more information on how are you going to process the data and what type of outcome will return your study.
Round 2
Reviewer 1 Report
I have no further concerns. I think the paper is now ready for publication